# Anti-Tumor Functions of Prelatent Antithrombin on Glioblastoma Multiforme Cells

**DOI:** 10.3390/biomedicines9050523

**Published:** 2021-05-07

**Authors:** Julia Peñas-Martínez, Ginés Luengo-Gil, Salvador Espín, Nataliya Bohdan, Carmen Ortega-Sabater, Maria Carmen Ródenas, David Zaragoza-Huesca, María José López-Andreo, Carme Plasencia, Vicente Vicente, Alberto Carmona-Bayonas, Irene Martínez-Martínez

**Affiliations:** 1Servicio de Hematología y Oncología Médica, Hospital Universitario Morales Meseguer, Centro Regional de Hemodonación, Universidad de Murcia, IMIB-Arrixaca, 30003 Murcia, Spain; julia.penas@um.es (J.P.-M.); gines.luengo@um.es (G.L.-G.); salvaalmudema@gmail.com (S.E.); nataliabhn@hotmail.es (N.B.); carmenorsab@gmail.com (C.O.-S.); mariacarmen.rodenas1@um.es (M.C.R.); davidzaragozahuesca5369@gmail.com (D.Z.-H.); vicente.vicente@carm.es (V.V.); 2Sección de Biología Molecular, El Área Científica y Técnica de Investigación (ACTI), Universidad de Murcia, 30003 Murcia, Spain; majoloan@um.es; 3Applied Research Using Omic Sciences S.L., 08028 Barcelona, Spain; carme.plasencia@aeromics.es; 4Centro de Investigación Biomédica en Red de Enfermedades Raras, U-765-CIBERER, Instituto de Salud Carlos III (ISCIII), 28029 Madrid, Spain

**Keywords:** angiogenesis, enteropeptidase, hepsin, prelatent antithrombin, glioblastoma multiforme, invasion, migration, STAT3, VEGFA

## Abstract

Antithrombin, the main physiological inhibitor of the coagulation cascade, exerts anti-tumor effects on glioblastoma multiforme cells. Antithrombin has different conformations: native, heparin-activated, prelatent, latent, and cleaved. The prelatent form has an intermediate affinity between latent and native antithrombin, although it is the most antiangiogenic form. Herein, we investigate the effect of this conformation on the tumorigenic processes of glioblastoma multiforme cells. Antithrombin forms were purified by chromatography. Chromogenic/fluorogenic assays were carried out to evaluate enteropeptidase and hepsin inhibition, two serine proteases involved in these processes. Wound healing, Matrigel invasion and BrdU incorporation assays were performed to study migration, invasion and proliferation. E-cadherin, Vimentin, VEGFA, pAKT, STAT3, pSTAT3, and pERK1/2 expression was assessed by Western blot and/or qRT-PCR. Prelatent antithrombin inhibited both enteropeptidase and hepsin, although it was less efficient than the native conformation. Exposure to prelatent antithrombin significantly reduced migration and invasion but not proliferation of U-87 MG, being the conformation most efficient on migration. Prelatent antithrombin down-regulated VEGFA, pSTAT3, and pERK1/2 expression in U-87 MG cells. Our work elucidates that prelatent antithrombin has surprisingly versatile anti-tumor properties in U-87 MG glioblastoma multiforme cells. This associates with resistance pathway activation, the decreased expression of tumorigenic proteins, and increased angiogenesis, postulating the existence of a new, formerly unknown receptor with potential therapeutic implications.

## 1. Introduction

Glioblastoma multiforme (GB) is the most aggressive and lethal primary malignant tumor of the central nervous system in adults [1]. Despite aggressive multi-modality treatments (surgery, radiotherapy, chemotherapy (temozolomide)), the median overall survival remains within the range of 12–15 months since diagnosis and has failed to increase in recent years [2,3]. This dismal patient outcome is largely attributed to the presence of highly infiltrative glial cells together with rich vascular proliferation, rapid tumor growth, intratumoral heterogeneity, and the emergence of resistance pathways, among other factors [2]. Given the lack of recent therapeutic advances, the search for new treatment targets and clarification of the complex molecular networks that govern tumor progression and treatment resistance become all the more pressing [4].

Antithrombin (AT) is a glycoprotein belonging to the serpin (serine protease inhibitors) superfamily; it is synthesized largely in the liver and secreted to plasma. It plays a key role in hemostasis, since it is the main physiological inhibitor of the coagulation cascade. Beyond its crucial role in hemostasis, AT has exhibited anti-angiogenic [5,6,7], anti-inflammatory [8,9,10], anti-apoptotic [11], and anti-viral properties [12]. Recently, our group has demonstrated that AT can also exert an anti-tumor effect by inhibiting enteropeptidase (EP) [13]. However, other serine proteases could fall under the control of AT or of its different forms. One example is hepsin, which belongs to the transmembrane type II serine protease superfamily and whose expression has been related with different tumors such as prostate, cervix or gastric cancer [14,15]. AT inherits the structural flexibility of the serpins that enables it to carry out its inhibitory function on its target proteases [16]. This functional flexibility denotes the capacity to adopt different three-dimensional conformations: native, heparin-activated, prelatent, latent, and cleaved. The native conformation is thermodynamically unstable and has poor inhibitory capacity. When native AT binds to its heparin cofactor, it triggers a conformational change that accelerates the inhibition of target proteases by ~1000-times, because the reactive loop becomes more accessible. In contrast, the latent and cleaved conformations are regarded as hyperstable proteins that lack inhibitory ability, due to the reactive loop being internalizing inside the central β-sheet [17]. Latent conformation is adopted when protein becomes senescent or, in vitro, is at elevated temperatures, which confers it with structural stability. In contrast, the cleaved form arises from a feedback mechanism associated with the inefficient inhibition of its target proteases. Despite not exhibiting the inhibitory capacity of native AT, latent and cleaved conformations have proven to be anti-angiogenic [7].

The prelatent form is particularly interesting, in as much as it is the most anti-angiogenic form of the four, despite the fact that it retains a certain inhibitory capacity, with an affinity for heparin mid-way between the latent and native conformations [6,18]. This profile makes its role in tumor cell programs especially intriguing. Against this backdrop, we have researched the effect of prelatent AT in tumorigenesis processes of U-87 MG glioblastoma–astrocytoma cells.

## 2. Materials and Methods

### 2.1. Cell Lines and Culture Media

Human GB cell lines U-87 MG and U-251 MG were purchased from American Type Culture Collection and European Collection for Authenticated Cell Cultures, respectively. The U-87 MG cell line from ATCC was authenticated by Bioidentity (Elche, Spain) by STR, according to the American National Standards Institute. STR profiling was carried out following the ANSI/ATCC ASN-0002-2011 guidance, Authentication of Human Cell Lines: Standardization of STR Profiling. For the U-251 MG cell line, manual DNA extraction was performed and the PowerPlex^®^ 16 System (Promega, Madrid, Spain) was used for STR analysis. STR profiles were analyzed by GeneMapper 5 software and the cellosaurus STR database (CLASTR) search tool of the Cellosaurus database (ExPASy) (https://web.expasy.org/cellosaurus/, accessed on 13 February 2020).

U-87 MG and U-251 MG cells were treated with 10 µg/mL cyclins (BM Cyclin, Sigma-Aldrich, Madrid, Spain) for the decontamination of possible mycoplasmas, and grown in DMEM medium containing 1 g/L or 4.5 g/L glucose, respectively (Gibco Thermo Fisher, Madrid, Spain) and both supplemented with 10% fetal bovine serum (Gibco Thermo Fisher, Spain), 1% sodium pyruvate (Gibco Thermo Fisher, Spain), 1% non-essential amino acids (Gibco Thermo Fisher, Spain), 1% GlutaMAX-L (Gibco Thermo Fisher, Spain), and 0.1% gentamicin (Gibco Thermo Fisher, Spain). Cells were maintained in T-75 tissue culture flasks (Fisher scientific, Madrid, Spain) and grown in 5% CO_2_ at 37 °C in a humidified incubator. 

### 2.2. Native and Prelatent AT Purification

Native AT was purified as previously described [19]. Briefly, α-AT was purified from a pool of human plasma from 4 healthy subjects by heparin affinity chromatography using HiTrap Heparin columns (GE Healthcare, Barcelona, Spain) and an ÄKTA Purifier (GE Healthcare, Barcelona, Spain). A gradient from 0 to 3 M NaCl in 50 mM Tris-HCl, pH 7.4 was applied for the elution and AT-containing fractions were later applied to a HiTrap Q column (GE Healthcare, Barcelona, Spain). Finally, proteins were eluted using a gradient from 0 to 1 M NaCl and desalted using dialysis tubing (Thermo Fisher, Spain). The purity of proteins was evaluated by silver staining of 8% SDS-PAGE, as indicated elsewhere [20]. Proteins were stored at −70 °C. In this study, we did not use β-AT isoform. Prelatent AT was purified using an ÄKTA Purifier, and slightly modifying the linear gradient previously described [18], as follows: 0.1–0.88 M NaCl gradient from 5–35 min, and hold at 0.88 M NaCl for 20 min. Finally, the prelatent AT obtained was dialyzed in 20 mM sodium phosphate, 20 mM NaCl, 0.1 mM ethylenediaminetetraacetic acid (EDTA), pH 7.4 buffer (buffer A), and concentrated in 1 mL using the Vivaspin^TM^ 20 system (Sartorius, Madrid, Spain); its concentration was determined. Additionally, buffer A was flowed through the heparin column and used as a control for cell experiments. 

### 2.3. EP and Hepsin Inhibition by Prelatent AT and Complex Formation

To elucidate the inhibitory mechanism on cell invasion, the interaction was studied between prelatent AT and two type II transmembrane serine proteases, EP and hepsin, involved in degrading the extracellular matrix. The EP inhibition assay was performed in a 96-well plate for 1 h, at 37 °C in a total volume of 100 µL as previously described [13]. Briefly, native or prelatent AT (0.5 µM), were incubated in the presence or absence of low molecular weight heparin (LMWH) (33 µM) (Bemiparin, Laboratorios farmacéuticos Rovi, Madrid, Spain) and EP (0.55 µM) (Sigma-Aldrich, Madrid, Spain), in TCNB buffer (50 mM Tris-HCl, 10 mM CaCl_2_, 150 mM NaCl, 0.05% Brij-35, pH 7.5). After adding the chromogenic substrate Z-Lys-SBzl (200 µM), the absorbance at 405 nm was measured on a microplate reader. Triplicate assays were performed for each condition.

Hepsin inhibition kinetics was also evaluated. Native and/or prelatent AT (0.05 µM) were incubated in the presence or absence of LMHW (Bemiparin, Laboratorios Farmacéuticos Rovi, Madrid, Spain) (33 µM) and hepsin (0.05 µM) (R&D Systems, Madrid, Spain) in 50 mM Tris-HCl, pH 9 buffer, for 1 h, at 37 °C. After the addition of the fluorogenic substrate BOC-Gln-Arg-Arg-AMC (200 µM) (Bachem, Barcelona, Spain), fluorescence emission was measured on a microplate reader, with excitation and emission wavelengths of 380 nm and 460 nm, respectively, for 5 min. Triplicate assays were performed for each condition.

The formation of covalent complexes was evaluated by 8% SDS-PAGE under non-reducing and reducing conditions, and Western blot, after incubating native or prelatent AT (0.17 µM) with EP (4.98 µM) (Sigma-Aldrich, Madrid, Spain) or hepsin (R&D Systems, Spain), for 1 hour, at 37 °C. Electrophoresis was conducted in reducing and non-reducing conditions, since it is known to affect the electrophoretic migration of cleaved AT. The interaction between AT and EP was also examined in the presence of LMHW (0.11 mM) (Bemiparin, Laboratorios Farmacéuticos Rovi, Madrid, Spain). For detection by Western blot, we used rabbit anti-human AT or anti-Hepsin polyclonal antibody (Cayman Chemical, Ann Arbor, MI, USA), followed in both cases by horseradish peroxidase-conjugated donkey anti-rabbit IgG, and detection using the ECL kit.

### 2.4. Wound Healing Assay

U-87 MG cells were cultured as confluent monolayers in a polystyrene microplate 6-well (Thermo Fisher Scientific, Madrid, Spain). The cells were then wounded by removing a 300–500 μm-wide strip of cells across the well with a standard 200 μL pipette tip and washed twice to remove non-adherent cells. The following six conditions were incubated with cells for 11 h: (1) control: supplemented with buffer A; (2) 3.88 µM native AT; (3) 2.16 μM prelatent AT; (4) 200 U/mL LMWH (enoxaparin, Sanofi-Aventis, Barcelona, Spain); (5) native AT and LMWH, (6) prelatent AT and LMWH. Finally, wound healing was quantified using Fiji-ImageJ software as the median percentage of the remaining cell-free area compared to the area of the initial wound. Triplicate assays were performed for each condition, and results were expressed relative to the control condition. 

### 2.5. Matrigel Invasion Assay

The experiment was performed at 37 °C for 6 h using 24-well transwell inserts coated with Matrigel (BD Biosciences, Madrid, Spain). Seventy thousand (70,000) U-87MG cells suspended in 400 μL of serum-free medium were seeded into the upper chamber and 500 μL of serum-supplemented medium was added in the lower chamber. The following six treatments were analyzed against untreated cells: (1) control (cells with serum-free medium); (2) 12.93 nM native AT; (3) 12.93 nM prelatent AT; (4) 200 U/mL LMWH (enoxaparin, clexane, Sanofi-Aventis, Barcelona, Spain); (5) native AT and LMWH, and (6) prelatent AT and LMWH. Cells that migrated and invaded the membrane were counted after fixing with methanol and staining with crystal violet. Images were analyzed with Fiji-ImageJ 1.52i software.

### 2.6. Cell Proliferation Assay

To evaluate the proliferative capacity of U-87 MG and U-251 MG cell lines under the effect of prelatent AT (2.16 µM) or buffer A, 8 × 10^5^ cells were grown in a polystyrene microplate 6-well (*n* = 3/group). After 11 h of treatment, cells were incubated with 10 µM 5-Bromo-20-deoxyuridine (BrdU) (552598; BD Biosciences, Madrid, Spain) for 24 h and 4 h, respectively. Then, the APC BD BrdU flow kit (552598, BD Biosciences) was used to fix and permeabilize cells prior to DNAse treatment and staining with anti-BrdU-APC. Finally, the incorporation of BrdU was detected by flow cytometry using a BD Accuri C6 flow cytometer device (Ann Arbor, MI, USA).

### 2.7. Real Time-PCR and Immunoblotting of Selected Cancer Signaling Proteins

U-87 MG cells were incubated with prelatent AT (2.16 µM), native AT (2.16 µM), or in buffer A (*n* = 4/group). After 11 h, total RNA was isolated using Trizol^®^ Reagent (Invitrogen, Carlsbad, CA, USA) following manufacturer’s instructions. The RNA concentration and 260/280 ratio were determined by using a NanoDrop spectrophotometer (Thermo Scientific, Wilmington, DE, USA). From total RNA, a 200-ng sample was reverse-transcribed to cDNA according to the manufacturer’s instructions (SuperScript First Strand, Invitrogen, Madrid, Spain). PCR reactions were carried out using TaqMan^®^ Gene Expression probes (E-cadherin (CDH1): hs01023894_m1; vascular endothelial growth factor A (VEGFA): hs00900055_m1; vimentin (VIM): hs00185584_m1; signal transducer and activator of transcription 3 (STAT3): hs00374280_m1) on a LC480 real-time PCR (Roche, Madrid, Spain) in triplicate for each sample. GAPDH (hs99999905_m1) expression was used as the endogenous reference control as its expression was not affected by the prelatent AT treatment. The fold difference for each sample was obtained using the 2-ΔCt method.

Concurrently, cells were lysed and samples separated by 8% SDS-PAGE under reducing conditions to evaluate pAKT, pERK1/2, pSTAT3, and VEGFA protein expression. Bands were detected by Western blot following standard immunoblotting procedures using polyvinylidene difluoride membrane. Primary anti-human antibodies (1:1000 dilution) were pAKT (Invitrogen, Madrid, Spain), pERK1/2 (9101S, Cell Signaling Technologies, Werfen, Barcelona, Spain), pSTAT3^(Tyr705)^ (9145, Cell Signaling Technologies, Werfen, Barcelona, Spain), and VEGFA (ab46154, Abcam, Barcelona, Spain) made in rabbit. Secondary IgG antibodies were horseradish peroxidase-coupled and visualized by the ECL kit detection. Protein expression of GAPDH (3683, Cell Signaling Technologies, Werfen, Barcelona, Spain) was used as the endogenous reference control. 

### 2.8. Statistics

The Kruskal–Wallis test was applied to evaluate genetic expression and percentage data according to different AT conformations. Pairwise comparisons were evaluated by means of post hoc Dunn tests adjusted with the Benjamini–Hochberg method.

## 3. Results

### 3.1. Inhibitory Effect of Prelatent AT on EP and Hepsin

We began by confirming the previously known inhibitory profile of the prelatent AT and its affinity for heparin, lower than native AT [18] (Appendix A). Subsequently, we studied the inhibitory effect of the prelatent AT on EP. As shown in Figure 1a, under reducing conditions, covalent complexes formation was not observed with any of the AT conformations, since bands around 90 kDa were not detected, which would be the estimated molecular weight for the complex between AT and EP. Meanwhile, under non-reducing conditions (Figure 1b), a cleaved AT band was observed in all reactions, with less migration than intact AT. These results with prelatent AT are consistent with those reported by our group for native AT in the past [13]. This observation is compatible with a mutual interaction consisting of the excision of prelatent AT (which becomes cleaved AT) and EP inhibition, without forming a covalent complex in any case. Therefore, we sought to assess whether EP modulated by prelatent AT continued to exhibit residual enzymatic activity (see Methods). What we observed was that the prelatent AT inhibited EP activity slightly, with a discreet enhancement of the inhibition (1.2 times) after LMWH binding. Meanwhile, in native AT, LMWH activation increased EP inhibition fivefold (Table 1) [13].

As hepsin belongs to the transmembrane type II serine protease superfamily and has also been related with different malignant tumors [14,15], we performed a similar study to evaluate the interaction between hepsin and native or prelatent AT. Unlike the previous case, the results point toward native AT being capable of establishing a covalent complex with hepsin under both reducing and non-reducing conditions. In contrast, this was not seen when the serine protease was incubated with the prelatent conformation (Figure 2a,b). Finally, residual enzymatic activity of hepsin was evaluated by fluorogenic measurement after the above-detailed incubations. As shown in Table 2, native AT was able to efficiently inhibit hepsin activity and, in the presence of LMHW, the inhibitory effect was slightly improved. However, prelatent AT scarcely inhibited the action of this serine protease.

Remarkably, inhibition was much greater when prelatent AT was incubated first with hepsin and then with native AT versus exposure to native AT alone, suggesting a change in the three-dimensional conformation resulting from the interaction. Conversely and surprisingly, when both AT conformations were incubated together in the presence of hepsin, the inhibition of the activity on the latter was equivalent to that obtained with the native conformation alone.

### 3.2. Prelatent AT Reduces Migration and Invasion of U-87 MG Cells

Next, we investigated the effect of native or prelatent AT on U-87 MG cell migration in the presence and absence of low molecular weight heparin (LMWH), using an in vitro wound healing assay. The wound confluence test was therefore performed (see Methods). As shown in Figure 3, prelatent AT was able to reduce cell migration substantially compared to the control (mean percentage of confluence, 55 vs. 72%, *p*-value < 0.001), and compared to native AT (mean wound confluence, 55 vs. 68%, *p*-value < 0.01), while native AT and the control revealed no differences (68 vs. 72%, respectively, *p* = 0.326).

This inhibition has a dose–response effect for prelatent AT (Appendix A). Thus, the prelatent conformation exerts its maximum effect on cell invasion at a concentration of 2.84 μM, which accounts for 72% of the maximum inhibitory concentration for native AT. The prelatent heparin complex did not improve the effect on these processes, as opposed to native AT which did exert a greater inhibitory effect when it reacted with its cofactor.

We also evaluated the effect of AT on invasion by a Matrigel invasion assay. The data are in keeping with a comparable decrease in invasiveness of GB cells when said cells are treated with native or prelatent AT, whether or not they are incubated with LMWH (Figure 4).

### 3.3. Prelatent AT Did Not Affect the Proliferation of U-87 MG or U-251 MG Cells

Glioma tumors have a highly proliferative phenotype, principally due to the loss of multiple cell-cycle inhibitors and to increased signaling from multiple growth factor receptors that positively regulates the cell cycle [21]. Therefore, to determine if prelatent AT could have an effect on proliferation processes, BrdU incorporation assay was evaluated by flow cytometry after 11 h of treatment. Unlike on migration and invasion processes, prelatent treatment had no significant effect on U-87 MG nor U-251 MG proliferation (Appendix A).

### 3.4. Prelatent AT Downregulates the Expression or Function of Different Cancer Signaling Molecules

Given the potential effect of prelatent AT on tumor processes, especially on cell migration, we examined whether treatment with this AT conformation altered U-87 MG gene expression. Epithelial–mesenchymal transition was gauged by measuring *CDH1* (E-cadherin gene) and *VIM* (vimentin gene) expression. Our results revealed that prelatent AT treatment did not alter the expression of either *CDH1* of VIM in U-87 MG cells (Appendix A). Since prelatent AT has already been proven to be anti-angiogenic and GB is characterized by high vascularization, we probed the effect of prelatent AT on VEGFA expression in U-87 MG cells and in comparison to native AT. As exhibited in Figure 5a and Figure 6a, prelatent AT treatment significantly downregulated VEGFA expression.

We also looked at the expression of STAT3 and pSTAT3, pERK1/2 and pAKT whose expression is crucial in key cancer signaling pathways and that have become potential targets for treated GB [22,23,24]. As shown in Figure 5b and Figure 6b, prelatent AT treatment significantly downregulated the pSTAT3-α expression, which is the isoform that can act as an oncogene [25]. On the other hand, prelatent AT reduced ERK1/2 phosphorylation, which was not attributable to crosstalk with AKT, whose activation was unaffected (Figure 5b).

Finally, we also evaluated the effect of prelatent AT treatment on a different GB cell line. However, treatment of U-251 MG cells with prelatent AT did not significantly modified the expression of these signal transduction factors (Appendix A), as it happened with U-87 MG cells.

## 4. Discussion

GB is the most common malignant brain tumor in adults and entails a poor prognosis across the board. Despite intense research, little progress has been made in recent years [22]. This involves the need to dilucidate novel mechanisms that feed the next generation of clinical trials. 

AT plays a key role in coagulation, but also has other functions beyond hemostasis [5,9,11,12]. Our work confirms that prelatent AT, which is physiologically present in our blood at very low levels, has multiple pleiotropic effects on U-87 MG cells, including a potent anti-angiogenic function, being capable of down-regulating VEGFA and STAT3 expression, which normally forms a transcriptional up-regulation circuit [6,26]. Concomitantly, prelatent AT is capable of lowering ERK1/2activation, which has been implicated in promoting tumor progression and mediates resistance to anti-tumor therapies [23]. This appears to be unmediated by the AKT pathway. Together, this array of capacities makes prelatent AT a potential partner in modulating anti-angiogenic treatment strategies, by enhancing them or delaying mechanisms of evasive resistance. 

Microvascular proliferation comprises one of the hallmarks of GB, as opposed to low grade glioma. Several factors have been implicated in generating the rich vascular network of these tumors [27]. Glial neo-angiogenesis is a highly complex tumor process with several interwoven steps; tumor hypoxia is one trigger, with the induction of hypoxia-inducible factor-1 (HIF-1) [28], expression of several associated transcription factors, and components of the extracellular matrix [29,30]. This leads to the imbalance between anti- and pro-angiogenic factors such as VEGF, transforming growth factor-β (TGF-β), epidermal growth factor (EGF), and others. These factors act like receptors in endothelial cells, which are recruited, which, in turn, entails the degradation of the endothelial cell basement membrane and extracellular matrix, together with different matrix metalloproteinases (MMPs) [31]. The process culminates in capillaries sprouting from preexisting blood vessels.

The transcriptional activation of the VEGFA gene in GB cells is one of the most conspicuous mechanisms involved [32]. In the phase III Avaglio randomized controlled trial (RCT), the addition of bevacizumab (a VEGFA-inhibiting monoclonal antibody) in combination with radiotherapy and chemotherapy, enhanced progression free-survival in patients with GB (10.6 months vs. 6.2 months; hazard ratio, 0.64; 95% confidence interval, 0.55 to 0.74; *p* < 0.001) [33]. Nevertheless, no differences were seen in overall survival, making the exploration of mechanisms to overcome GB resistance to anti-angiogenics of the utmost importance. Increased migration/invasion is a common mechanism of resistance, the clinical correlate of which is diffuse progression [34,35]. If validated in vivo, the co-repression shown here of the VEGFA and pSTAT3 signaling pathway in U-87 MG cells would endorse the role of prelatent AT in controlling resistance to anti-angiogenics.

The IL-6-STAT3 signaling pathway has already been recognized as one of the pathways involved in the progression of GB through feed-back loops that comprise both the hypoxia, as well as cytokines originating from the stroma [36,37]. Thus, STAT3 is part of a particularly interconnected circuit, with multiple roles in tumor progression, evasion of the immune response, anti-apoptotic activity, angiogenesis, etc. [38]. The co-option of reactive astrocytes of the tumor microenvironment by means of STAT3 signaling fosters the implantation and survival of metastasis by modulating the immune system [39]. Up-regulation of the pSTAT3 pathway has been correlated with the development of evasive resistance to bevacizumab in individuals with GB [40]. STAT3 inhibition by compounds such as silybilin has been put forth as an effective option [41]. However, there is currently only one active phase I study that is assessing a STAT3inhibitor in GB (NCT01904123).

In prelatent AT-treated U-87 MG cells, this activity on pSTAT3 could be coordinated with ERK1/2 inactivation by dephosphorylation. The MEK-ERK1/2 pathway exhibits aberrant activity in 90% of all GB [42], which represents a mechanism of evasive resistance during EGRF inhibitor therapy [43]. In addition, this pathway correlates with increased invasion and GB cell migration [44]. These actions require that the extracellular matrix be degraded by matrix metalloproteinases (MMPs) and several components of the hepsin/transmembrane protease, serine (TMPRSS) subfamily [15,45,46,47,48]. 

We have proven the inhibitory effect of heparin-activated native and prelatent AT on cell migration and invasion of U-87 MG cells. This mechanism can be explained, at least in part, by the inhibition of the type II transmembrane serine protease EP [13]. It is likely that other proteases participate in the degradation of the extracellular matrix. Furthermore, the connection via MEK–ERK mediates part of this process of resistance to antiangiogenic therapies [44]. However, MEK inhibition with trametinib provides a limited effect, considering that proliferation control is rapidly compensated for by the acquisition of more aggressive phenotypes [49]. Effective handling of this complex network requires that several nodes be inhibited simultaneously [24]. 

As for applicability, prelatent AT maintains residual anticoagulant activity, making it possible to foresee clinical ramifications in the field of hemostasis. Thromboembolic disease is common in patients with GB, having a long-term accumulated incidence of up to 28% [50] and no effective thromboprophylactic strategy [51]. Bevacizumab slightly increases the risk of both thrombosis and bleeding [52,53]. Given that prelatent AT represses VEGFA while retaining its anticoagulant capacity, its in vivo interaction with bevacizumab should be the subject of a separate analysis in the future. To do so demands the detailed characterization of prelatent AT receptors on the glial cell, as well as the precise and currently unknown mechanism of the inactivation of the EGFR pathway. Likewise, inhibitory drugs targeting this pathway with modulable anticoagulant activity must be designed.

Our study has several limitations, the most notable of which is inherent in general cell line studies, which does not make it possible to replicate the architecture, spatial heterogeneity, and interactions with the stroma of the tumor. Moreover, U-87 MG cells have been assimilated into the mesenchymal subtype of The Cancer Genome Atlas Network (TCGA) classification, despite exhibiting differences in some proteases and mesenchymal genes [54,55]. Additionally, prelatent AT treatment did not have the same effect on the U-251 MG cell line. The explanation to the different behavior of U-87 and U-251 MG cells under treatment with prelatent AT could be the differences in protein expression between the two cells lines, as it has been previously reported [56]. In fact, although it is only a speculation, the potential receptor of prelatent AT in U-87 MG cells could be absent in U-251 MG cells. Finally, these findings must be validated on real samples covering the entire molecular spectrum of GB (e.g., IDH 1/2 status) and EGFR’s mutational profile.

In summary, we have described a new and surprisingly versatile mechanism by the means of which a special conformation of AT subjected to stress (prelatent AT) is able to take on antitumor abilities at several levels on U-87 MG cells. Not least among these capabilities is the inhibition of the processes of invasion, migration, angiogenesis, and treatment resistance through the modulation of critical signaling pathways in glial cells, such as VEGFA, ERK1/2 and STAT3.Therefore, the anti-tumor effect of prelatent AT should be explored in the treatment of GB where prognosis is poor given the limited efficacy of conventional therapy.

## Figures and Tables

**Figure 1 biomedicines-09-00523-f001:**
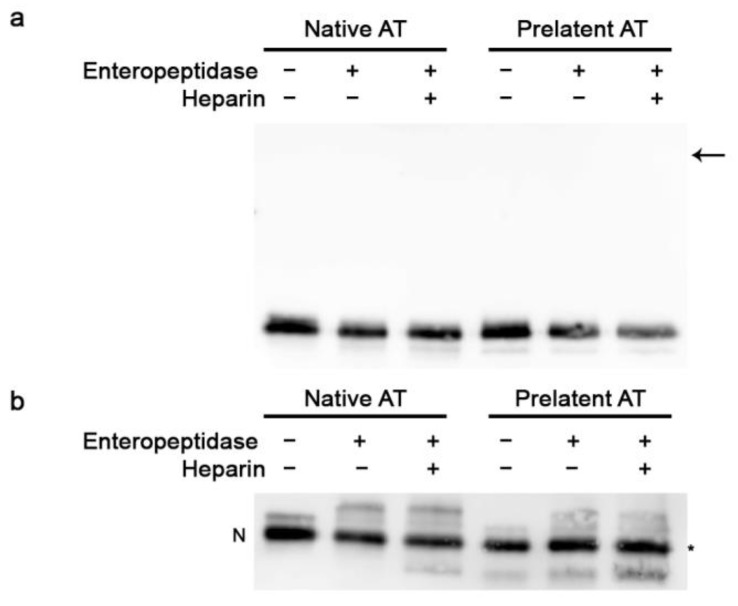
Electrophoretic evaluation of the interaction between antithrombin and enteropeptidase. SDS-PAGE was run under reducing (+dithiothreitol (DTT)) (**a**) and non-reducing (−DTT) conditions (**b**), and Western blotting with immunodetection with anti-antithrombin antibody was performed. The asterisk represents cleaved antithrombin. N indicates the electrophoretic mobility in SDS for antithrombin in its native conformation. The arrow represents the covalent complex expected for antithrombin–enteropeptidase (~90 kDa). AT: antithrombin.

**Figure 2 biomedicines-09-00523-f002:**
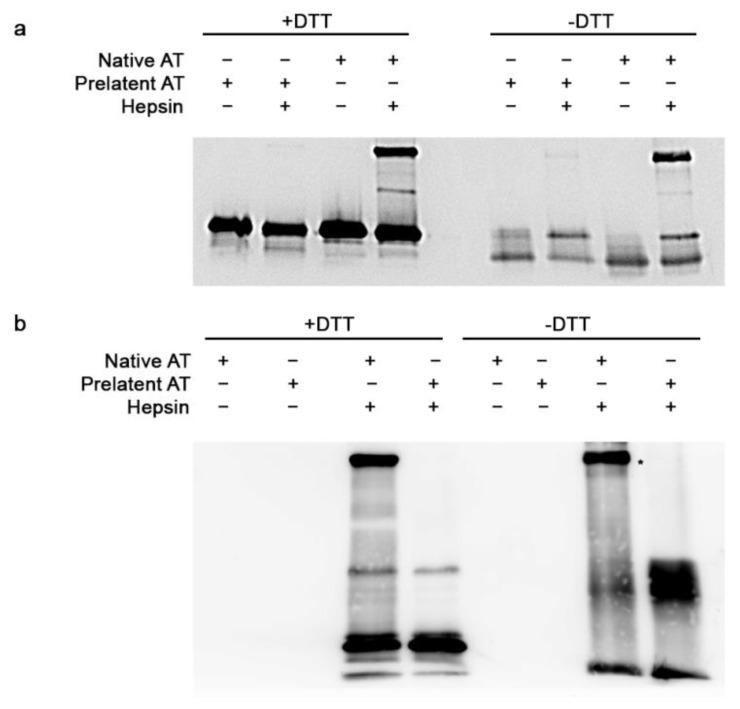
Electrophoretic evaluation of the interaction between antithrombin and hepsin. SDS-PAGE was run under reducing (+DTT) (**a**) and non-reducing (−DTT) conditions (**b**), and Western blotting with immunodetection with anti-antithrombin antibody (**a**) or anti-hepsin antibody (**b**) was performed. Asterisk represents antithrombin–hepsin complex. AT: antithrombin.

**Figure 3 biomedicines-09-00523-f003:**
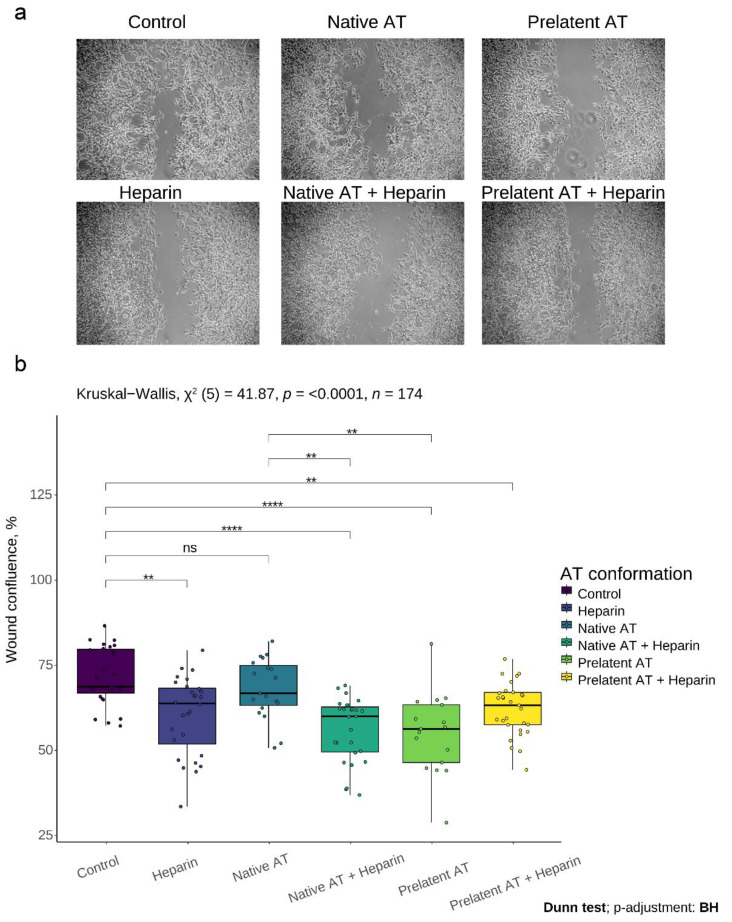
Effects of antithrombin on cell migration of U-87 MG cells. Wound healing was evaluated after incubation of cells for 16 h with no treatment (control), low molecular weight heparin (LMWH), native antithrombin (AT), or prelatent AT, and each AT in combination with LMWH. (**a**) Microscope images of cells 16 h after the wound created with the pipette tip. (**b**) Percentage of wound confluence under the different conditions. Each condition was evaluated in triplicate, and 5 different images were processed for each assay; ** *p* < 0.01; **** *p* < 0.0001; ns = not significant. Images were recorded with a Leica microscope at 5× and Fiji ImageJ was used to analyze migration.

**Figure 4 biomedicines-09-00523-f004:**
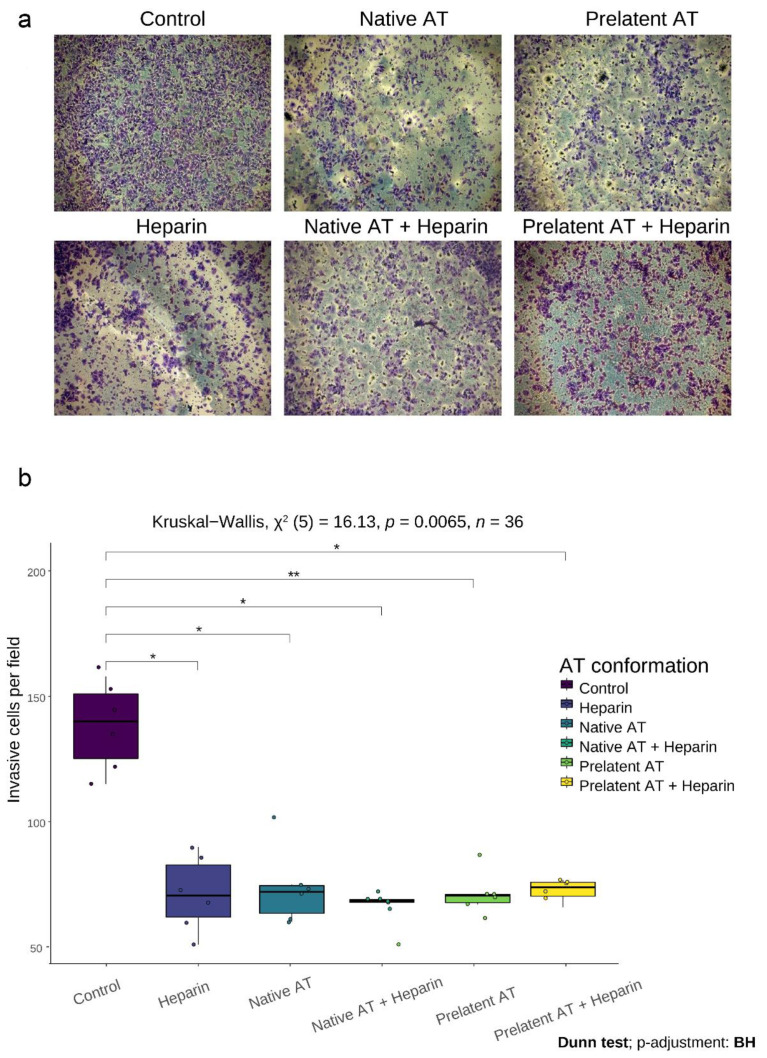
Effects of antithrombin on invasion by transwell assay. Cell invasion was evaluated after incubation of U-87 MG cells for 6 h with no treatment (control), low molecular weight heparin (LMWH), native antithrombin (AT), or prelatent AT, and each AT in combination with LMWH. (**a**) Microscope images of cells invaded after 6 h of incubation. (**b**) Percentage of cells invading under the different conditions. Each condition was evaluated in triplicate, and 3 different images were processed for each assay; * *p* < 0.05; ** *p* < 0.01. Images were recorded with a Leica microscope at 5×, and Fiji-ImageJ was used to analyze invasion. Abbreviations: BH = Benjamini–Hochberg procedure; AT = antithrombin.

**Figure 5 biomedicines-09-00523-f005:**
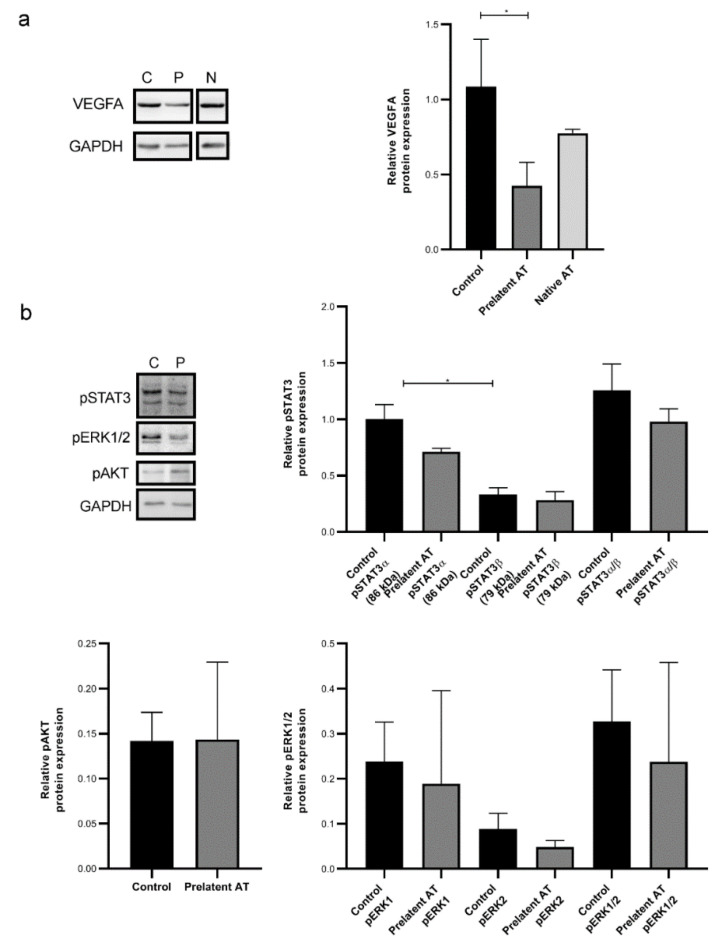
VEGFA, pSTAT3, pAKT, and pERK1/2 expression on U-87 MG cells. (**a**) Electrophoresis and Western blot of VEGFA in lysates of U-87 MG cells treated with buffer (C), prelatent (P) or native antithrombin (N). GAPDH expression was detected as loading control. (**b**) Electrophoresis and Western blot of pSTAT3, pERK1/2 and pAKT in lysates of U-87 MG cells treated with buffer (C) or prelatent antithrombin (P). GAPDH expression was detected as loading control. AT: antithrombin. * *p* < 0.05.

**Figure 6 biomedicines-09-00523-f006:**
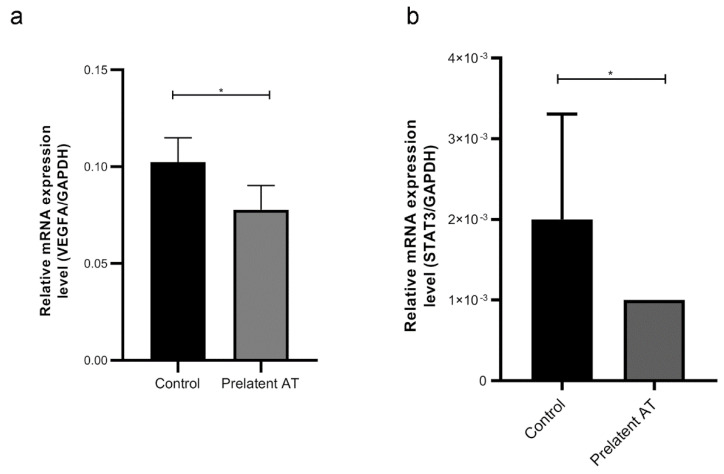
Relative expression of *VEGFA* (**a**), and *STAT3* (**b**) to *GAPDH* mRNA (2-ΔCt). Each condition was evaluated in triplicate, and 5 different samples per group were analyzed. * *p* < 0.05.

**Table 1 biomedicines-09-00523-t001:** Enteropeptidase activity using its chromogenic substrate after incubation with antithrombin.

Condition	Vmax_405_ (mOD/min)
Enteropeptidase	3.357 ± 0.311
Native antithrombin + Enteropeptidase	2.714 ± 0.679
Prelatent antithrombin + Enteropeptidase	2.976 ± 1.209
Native antithrombin + Enteropeptidase + Heparin	0.536 ± 0.258
Prelatent antithrombin + Enteropeptidase + Heparin	2.571 ± 0.619

Enteropeptidase activity was evaluated after incubation of native or prelatent antithrombin in the presence or absence of low molecular weight heparin. The values are represented as the mean of 3 independent experiments.

**Table 2 biomedicines-09-00523-t002:** Hepsin activity using its fluorogenic substrate after incubation with antithrombin.

Condition	Vmax_405_ (mOD/min)
Hepsin	3.67 × 10^6^ ± 4.69 × 10^5^
Native antithrombin + Hepsin	3.21 × 10^6^ ± 1.81 × 10^5^
Native antithrombin+ Heparin + Hepsin	2.38 × 10^6^ ± 1.91 × 10^5^
Prelatent antithrombin+ Hepsin	3.65 × 10^6^ ± 1.15 × 10^5^
Prelatent antithrombin+ Hepsin + Native antithrombin	3.15 × 10^6^ ± 2.86 × 10^5^
* Prelatent antithrombin+ Hepsin + Native antithrombin	1.19 × 10^6^ ± 3.15 × 10^5^

Hepsin activity was evaluated after incubation of native or prelatent antithrombin in the presence or absence of low molecular weight heparin. Values are represented as the mean of 3 independent experiments. The asterisk indicates that prelatent antithrombin was incubated first with hepsin and then with native antithrombin.

## Data Availability

Not applicable.

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
