# Peer review of "Anti-Tumor Functions of Prelatent Antithrombin on Glioblastoma Multiforme Cells"

_biomedicines, 2021, doi:10.3390/biomedicines9050523_

Round 1
Reviewer 1 Report
The manuscript of Julia Peñas-Martínez et al. Article Title: Anti-tumor functions of prelatent antithrombin on glioblastoma multiforme cells. Journal: Biomedicines Manuscript no. 1142158
is interesting and well prepared. I believe that the conducted research can contribute a lot to the state of knowledge in the field of molecular biology of glioblastoma. The research model is well developed and the content of the work itself requires only minor changes as listed below.
P.42 Instead of (surgery, radiotherapy, temozolomide), I propose (surgery, radiotherapy, chemotherapy (temozolomide)).
- 42 Can the average survival value also be given?
- 86 Why do you use a medium other than that recommended by the manufacturer?
- 89 Why are you taking an antibiotic other than the usual one?
- 137 Please explain the abbreviation of the putty.
- 171 Why are you using GAPDH as a reference gene, please explain.
- 217-219 “As hepsin belongs to the transmembrane type II serine protease superfamily and has also been related with different malignant tumors [19,20], we performed a similar study to evaluate the interaction between hepsin and native or prelatent AT " I would like to see this sentence in introdaction and elaborate on it a bit.
- 249-250 Please indicate in the sentence that the result was not statistically significant except for the p-value itself.
- 285 Please add why you did not include this data or include them.
Author Response
The manuscript of Julia Peñas-Martínez et al. Article Title: Anti-tumor functions of prelatent antithrombin on glioblastoma multiforme cells. Journal: Biomedicines Manuscript no. 1142158 is interesting and well prepared. I believe that the conducted research can contribute a lot to the state of knowledge in the field of molecular biology of glioblastoma. The research model is well developed and the content of the work itself requires only minor changes as listed below.
We would like to thank the reviewer for the kindly words and for giving us the opportunity to improve our research with his/her suggestions.
P.42 Instead of (surgery, radiotherapy, temozolomide), I propose (surgery, radiotherapy, chemotherapy (temozolomide)).
We have rewritten this sentence as suggested by the reviewer (line 47).
P.42 Can the average survival value also be given?
Although the average survival time can provide useful information (e.g., restricted mean survival times), this measure of central tendency is not usually reported in oncology clinical trials. The reason is that the survival function is often skewed to the right, so the mean is frequently influenced by outliers, whereas the median is a more robust estimator. Furthermore, the median can be estimated consistently in the presence of censored survival times. To our knowledge the average survival time for high-grade gliomas has been estimated only in the context of retrospective studies, without very substantial insights with respect to the median (Kushnir I, Tzuk-Shina T. Isr Med Assoc J. 2011 May;13(5):290-4. https://pubmed.ncbi.nlm.nih.gov/21845970/)
Therefore, we believe that adding the average does not provide incremental information.
P.86 Why do you use a medium other than that recommended by the manufacturer?
Although in the ATCC EMEM medium is indicated, it has been extensively reported that U-87 MG cells also grow in DMEM medium (Oh S-J. et al. Cancer Cell Int. 2017 Feb;17:22. https://pubmed.ncbi.nlm.nih.gov/28203118/, Shin S.Y. et al. Carcinogenesis. 2013 Sep;34(9):2089-9. https://pubmed.ncbi.nlm.nih.gov/23689352/, Lu K. et al. Oncol Lett. 2018 Aug;16(2)2478-2482. https://pubmed.ncbi.nlm.nih.gov/30013640/, Banayoun L. et al. Cancer Biol Ther. 2013 Jan;14(1):64-74. https://pubmed.ncbi.nlm.nih.gov/23114641/)
P.89 Why are you taking an antibiotic other than the usual one?
Gentamicin has broad spectrum antibacterial and anti-mycoplasmal activity as well as free from cell toxicity. Moreover, it has biological and biochemical properties, which render it superior to the use of Penicillin and Streptomycin (PS). The main advantages are that as well as being active against many gram positive or gram negative organisms, gentamicin is active against strains of Proteus and Staphylococcus which can sometimes be resistant to a PS mix and unlike PS, gentamicin is active against Pseudomonas strains. Moreover, although not active against yeasts, moulds or protozoa, gentamicin has been reported to be active against several but not all strains of mycoplasma.
Thus, it has been reported its use in the culture medium of other cells, included cancer cells from the nervous systems (Durbas M. et al. Apoptosis. 2018 Oct;23(9-10):492-511. https://pubmed.ncbi.nlm.nih.gov/30027525/)
P.137 Please explain the abbreviation of the putty.
As indicated by the reviewer, we have described the abbreviation of low molecular weight heparin (New line 129-130).
P.171 Why are you using GAPDH as a reference gene, please explain.
We used this gene as we confirmed that the expression was not affected by the prelatent AT treatment as it happened with b-actin. We have incorporated this information in the manuscript (lines 197-198).
P.217-219 “As hepsin belongs to the transmembrane type II serine protease superfamily and has also been related with different malignant tumors [19,20], we performed a similar study to evaluate the interaction between hepsin and native or prelatent AT " I would like to see this sentence in introduction and elaborate on it a bit.
As suggested by the reviewer this information has been incorporated in the introduction. “One example is hepsin, which belongs to the transmembrane type II serine protease superfamily and whose expression has been related with different tumors such as prostate, cervix or gastric cancer (Cheng, H. et al. Oncol. Lett. 2017, 14, 159–164. https://pubmed.ncbi.nlm.nih.gov/28693148/, Zhang, M. et al. Nat. Publ. Gr. 2016, 6, 36902. https://pubmed.ncbi.nlm.nih.gov/27841306/). (New lines 62-64).
P.249-250 Please indicate in the sentence that the result was not statistically significant except for the p-value itself.
The reviewer has identified an imprecision in the analysis and we agree that the figure and this part of the text should be clarified. We have therefore uploaded a new figure 3 and rewritten the paragraph and clarified, following his/her recommendation as it follows: “As shown in Figure 3, prelatent AT was able to reduce cell migration substantially compared to control (mean percentage of confluence, 55 vs. 72%, p-value <0.001), and compared to native AT (mean wound confluence, 55 vs. 68%, p-value <0.01), while native AT and control revealed no differences (68 vs. 72%, respectively, p=0.326).” (New lines 299-302)
P.285 Please add why you did not include this data or include them.
As suggested by the reviewer this information has been incorporated as Supplementary Figure 6 (New line 422).

Reviewer 2 Report
This manuscript entitled “Anti-Tumor Functions of Prelatent Antithrombin on Glioblastoma Multiforme Cells” shows biochemical analyses of native and prelatent antithrombin and cellular analyses which reveal effects of antithrombin on glioblastoma cells. This manuscript is of value because the importance of post-translational events (protease processing and protein conformation changes) is confirmed, and one of the candidate clues for treatment of glioblastoma identified. However, this reviewer has two concerns that should be addressed before recommendation for publication.
(1) Effects of antithrombin on migration, invasion, and signal transduction factors were analyzed using only one cell line (U87 MG). At least, one or two more glioblastoma cell line(s) should be checked.
(2) In addition to migration and invasion, effects of antithrombin on proliferation (which can have an affect on migration and invasion assays) should be analyzed.
Author Response
This manuscript entitled “Anti-Tumor Functions of Prelatent Antithrombin on Glioblastoma Multiforme Cells” shows biochemical analyses of native and prelatent antithrombin and cellular analyses which reveal effects of antithrombin on glioblastoma cells. This manuscript is of value because the importance of post-translational events (protease processing and protein conformation changes) is confirmed, and one of the candidate clues for treatment of glioblastoma identified. However, this reviewer has two concerns that should be addressed before recommendation for publication.
We want to thank the reviewer for his/her kind words and for giving us the opportunity to improve our research with his/her suggestions.
- Effects of antithrombin on migration, invasion, and signal transduction factors were analyzed using only one cell line (U87 MG). At least, one or two more glioblastoma cell line(s) should be checked.
As suggested by the reviewer, we have checked some of our results with U-251 MG cell line. As discussed in the manuscript, the expression of VEGFA, ERK 1/2 and STAT3 is related with processes of invasion, migration, angiogenesis, and treatment resistance of cancer cells. However, treatment of U-251 MG cells with prelatent antithrombin did not significantly modified the expression of these signal transduction factors (Figure Supplementary 7), as it happened with U-87 MG cells. The explanation to the different behavior of U-87 and U-251 MG cells under treatment with prelatent antithrombin could be the differences in proteins expression between the two cells lines as it has been previously reported (Li, H. et al. Turk. Neurosurg. 2017, 27, 894–903, https://pubmed.ncbi.nlm.nih.gov/27651343/). In fact, although it is only a speculation, the potential receptor of prelatent antithrombin in U-87 MG cells could be absent in U-251 MG cells. These results and the discussion have been incorporated in the revised manuscript (New lines 88-90, 93-98, 441-444, 529-534).
- In addition to migration and invasion, effects of antithrombin on proliferation (which can have an effect on migration and invasion assays) should be analyzed.
We have checked the effect of prelatent antithrombin on proliferation as suggested by the reviewer. As shown in Supplementary Figure 5, prelatent antithrombin did not provoke any effect on proliferation and a similar effect was observed with U-251 MG cells. This information has been included in the revised manuscript (176-184, 388-395).

Round 2
Reviewer 2 Report
This manuscript has been improved.